# The Protective Effects on Ischemia–Reperfusion Injury Mechanisms of the Thoracic Aorta in Daurian Ground Squirrels (*Spermophilus dauricus*) over the Torpor–Arousal Cycle of Hibernation

**DOI:** 10.3390/ijms231810248

**Published:** 2022-09-06

**Authors:** Yuting Han, Weilan Miao, Ziwei Hao, Ning An, Yingyu Yang, Ziwen Zhang, Jiayu Chen, Kenneth B. Storey, Etienne Lefai, Hui Chang

**Affiliations:** 1Shaanxi Key Laboratory for Animal Conservation, Northwest University, Xi’an 710069, China; 2Key Laboratory of Resource Biology and Biotechnology in Western China, College of Life Sciences, Northwest University, Ministry of Education, 229# North Taibai Road, Xi’an 710069, China; 3Department of Biology, Carleton University, Ottawa, ON K1S 5B6, Canada; 4INRAE, Unité de Nutrition Humaine, Université Clermont Auvergne, UMR 1019, F-63000 Clermont-Ferrand, France

**Keywords:** torpor–arousal, ischemia–reperfusion, thoracic aortic, phenotypic switching, permeability

## Abstract

Hibernators are a natural model of vascular ischemia–reperfusion injury; however, the protective mechanisms involved in dealing with such an injury over the torpor–arousal cycle are unclear. The present study aimed to clarify the changes in the thoracic aorta and serum in summer-active (SA), late-torpor (LT) and interbout-arousal (IBA) Daurian ground squirrels (*Spermophilus dauricus*). The results show that total antioxidant capacity (TAC) was unchanged, but malondialdehyde (MDA), hydrogen peroxide (H_2_O_2_), interleukin-1β (IL-1β) and tumor necrosis factor α (TNFα) were significantly increased for the LT group, whereas the levels of superoxide dismutase (SOD) and interleukin-10 (IL-10) were significantly reduced in the LT group as compared with the SA group. Moreover, the levels of MDA and IL-1β were significantly reduced, whereas SOD and IL-10 were significantly increased in the IBA group as compared with the SA group. In addition, the lumen area of the thoracic aorta and the expression of the smooth muscle cells (SMCs) contractile marker protein 22α (SM22α) were significantly reduced, whereas the protein expression of the synthetic marker proteins osteopontin (OPN), vimentin (VIM) and proliferating cell nuclear antigen (PCNA) were significantly increased in the LT group as compared with the SA group. Furthermore, the smooth muscle layer of the thoracic aorta was significantly thickened, and PCNA protein expression was significantly reduced in the IBA group as compared with the SA group. The contractile marker proteins SM22α and synthetic marker protein VIM underwent significant localization changes in both LT and IBA groups, with localization of the contractile marker protein α-smooth muscle actin (αSMA) changing only in the IBA group as compared with the SA group. In tunica intima, the serum levels of heparin sulfate (HS) and syndecan-1 (Sy-1) in the LT group were significantly reduced, but the serum level of HS in the IBA group increased significantly as compared with the SA group. Protein expression and localization of endothelial nitric oxide synthase (eNOS) was unchanged in the three groups. In summary, the decrease in reactive oxygen species (ROS) and pro-inflammatory factors and increase in SOD and anti-inflammatory factors during the IBA period induced controlled phenotypic switching of thoracic aortic SMCs and restoration of endothelial permeability to resist ischemic and hypoxic injury during torpor of Daurian ground squirrels.

## 1. Introduction

Hibernation is a survival strategy that many animals have evolved to withstand harsh external natural environments [1]. During hibernation, animals undergo extended periods of torpor in which they do not eat or drink and remain inactive with body temperature falling to near ambient, interrupted only by short bouts of arousal [2]. Between the rapid alternation in torpor and arousal, physiological changes resemble those of ischemia–reperfusion; similar to ischemia, body temperature, blood flow, heart rate, cellular metabolic rate and energy expenditure are reduced during torpor [3,4,5], and similar to reperfusion, heart rate and oxygen consumption increase and body temperature rises during arousal [3,4,6].

In non-hibernator species, the restoration of blood flow to tissues after an ischemic event leads to both ischemic injury and reperfusion injury [7]. Ischemia leads to hypoxia [8] and inflammation [9,10], and reperfusion results in persistent or increased ischemic injury due to increased production of inflammatory factors and ROS [11,12,13]. By contrast, chronic ischemia and hypoxia result in vascular remodeling [14,15]. During remodeling, ROS promote vascular remodeling by inducing the growth and migration of vascular smooth muscle cells (VSMCs) [16]. In addition, ischemia–reperfusion promotes the release of TNFα, which in turn promotes the proliferation of VSMCs [17], ultimately leading to a phenotypic switch from contractile to synthetic VSMCs [18,19,20]. This was evident in the thickening of thoracic aortic mesothelium after 45 min of hypoxic–ischemia in rats [21] and in the thickening of the wall of small coronary arteries after 3 weeks of hypoxia in mice [13]. In addition, thickening of the pulmonary artery wall and narrowing of the lumen in rats after 28 days of low-pressure hypoxia [22] was accompanied by increased protein expression of the VSMC synthetic marker proteins, PCNA and OPN, and decreased protein expression of the contractile marker protein SM22α [23]. Reperfusion triggers further VSMC proliferation [24], and reperfusion after transient ischemia for 1 h is sufficient to induce a sustained smooth muscle cell phenotype switch within 28 days in mice [25]. However, one study found that IL-10 dose-dependently inhibits TNFα-induced smooth muscle cell phenotype conversion [17]. On the other hand, the release of TNFα induced by ischemia and reperfusion can also lead to (a) endothelial dysfunction by downregulating mRNA and protein expression of eNOS [26] and (b) increased vascular permeability by promoting interleukin 6 (IL-6) production, inducing dimerization of the endothelial cell (EC) surface receptor GP-130 and disrupting the tight junctions of ECs [27]. In addition, the same result is obtained with increased production of IL-1β [28]. It has also been noted that endothelial glycocalyx regulates vascular permeability [29]; however, both ischemia and reperfusion induce glycocalyx degradation under the influence of oxygen free radicals and xanthine oxidoreductase activity, which is detected as elevated HS [30,31,32,33]. Capillary endothelial permeability is increased in mouse brain after 14 days of chronic hypoxia [34]. Furthermore, vascular dysfunction and increased vascular permeability were found after placental ischemia in human infants [26]. In another study, renal vascular endothelial permeability was significantly increased in rat kidney during 48 h of reperfusion after 40 min of renal ischemia [35]. Thus, ischemia and ischemia–reperfusion can lead to dysregulation of systemic oxidative stress and inflammation levels in non-hibernators, thereby inducing phenotypic switching of VSMCs and increased endothelial dysfunction and permeability.

However, hibernating mammals exhibit a natural tolerance to ischemia during torpor as compared to non-hibernators [36]. For example, the expression of hypoxia-related proteins such as the hypoxia-inducible transcription factor-1α (HIF-1α) was significantly reduced in brain tissue in bats (*Hipposideros terasensis*) during hibernation [37], and woodchuck (*Marmota monax*) heart showed resistance to ischemia-induced arrhythmias and sudden cardiac death during torpor [38]. Moreover, increased H_2_S production and increased protein expression of smooth muscle actin, collagen and transforming growth factor have also been found in lungs of Syrian golden hamsters (*Mesocricetus auratus*) during hibernation, suggesting that hamster lungs undergo remodeling and the inflammatory response may be inhibited during torpor [39,40]. Interestingly, even in the non-hibernating months, Arctic ground squirrels (*Spermophillus parryii*) can tolerate at least 10 min of whole-brain ischemia without causing neuronal damage [41,42]. Furthermore, the upregulation of inflammatory cytokines during torpor can be completely eliminated during interbout arousals [43]. Periodic arousal was also found to contribute to the clearance of potentially toxic oxidative-stress-related metabolites in the plasma of hibernating thirteen-lined ground squirrel *(Ictidomys tridecemlineatus*) [44]. Interestingly, vascular remodeling and immune responses can modulate each other [45]. Although chronic responses may lead to permanent dysfunction and tissue fibrosis, moderate inflammation can stimulate angiogenesis and restore local homeostasis [46]. Vascular studies on hibernators have shown that brown bears (*Ursus arctos*) exhibit an inflammatory state during torpor, accompanied by a thicker mid-membrane in the aortic arch and a marked increase in VSMCs [47], but no deterioration in vascular function [48]. In addition, increased vascular endothelial permeability is found in hibernating thirteen-lined ground squirrels [49,50], and the endothelial dysfunction occurs in the carotid artery of Syrian golden hamsters during hibernation [51]. However, vascular-related changes in torpor are easily reversed during arousal without significantly impairing organ function [52]. Hence, previous studies have shown that large vascular vessels such as the aortic arch and carotid arteries have a natural resistance to ischemia–reperfusion injury in hibernators.

The thoracic aorta is located in the upper part of the descending aorta, the largest blood channel in the body, and has well-developed smooth muscle. However, little research has been carried out on thoracic aorta resistance to ischemia–reperfusion injury during the torpor–arousal cycle. Therefore, we hypothesized that the oxidative stress and systemic inflammation levels associated with the torpor–arousal cycle can induce controlled phenotypic switching of smooth muscles and alter endothelial permeability to assist in preventing/minimizing ischemia–reperfusion injury in Daurian ground squirrels. The present study examined changes in oxidative stress and inflammation levels, phenotypic switching and endothelial permeability of thoracic aortic smooth muscle, as well serum markers, in ground squirrels in sampled in different states: summer active (SA), late torpor (LT) and interbout arousal (IBA). The data further reveal the mechanisms of vascular resistance to ischemia–reperfusion injury in hibernators over the torpor—arousal cycle.

## 2. Results

### 2.1. Oxidative Stress Levels in Thoracic Aorta and Serum of Ground Squirrels

Compared with the SA group, the MDA levels in the thoracic aorta of LT ground squirrels were elevated by 118.8% (*p <* 0.01) (Figure 1A). H_2_O_2_ levels were elevated by 95.1% (*p <* 0.01) in the LT group and 80.3% (*p <* 0.01) in the IBA group (Figure 1B), respectively. Compared with the LT group, the MDA content was decreased by 58.8% (*p <* 0.01) in the IBA group (Figure 1A). In contrast, the TAC was maintained in both LT and IBA groups as compared with the SA group (Figure 1C). Furthermore, SOD levels in serum were decreased by 45.3% (*p <* 0.01) in the LT squirrels as compared with the SA group (Figure 1D), whereas SOD rose back towards the SA value in the IBA, rising by 57.5% (*p <* 0.05) over the LT value (Figure 1D).

### 2.2. Inflammatory Factor Levels in the Serum of Ground Squirrels

In terms of pro-inflammatory factors, the IL-1β level in serum increased by 47.8% (*p <* 0.01) in the LT group (Figure 2A) over the SA value, whereas the value dropped in IBA back towards the SA value, and the IL-1β level decreased by 20.2% (*p <* 0.05) in the IBA group as compared with the LT group but was still higher than in the SA group (Figure 2A). TNFα levels increased by 56.7% (*p <* 0.01) in the LT group and 41.8% (*p <* 0.05) in the IBA group (Figure 2B) as compared with the SA group. In terms of anti-inflammatory factors, the IL-10 level in the serum was reduced by 20% (*p <* 0.05) in LT ground squirrels as compared with the SA group but rose again in the IBA group (Figure 2C), whereas the IL-4 level showed no change between LT and IBA groups (Figure 2D). Besides, the IL-10 level was increased by 31.6% (*p <* 0.01) in the LT group as compared with the IBA group (*p <* 0.01, Figure 2C).

### 2.3. Structural Changes in SMC Phenotype in the Thoracic Aorta of Ground Squirrels

The thickness of the SMC layer was significantly increased by 37% (*p <* 0.01) and 32.2% (*p <* 0.01), respectively, in the IBA group as compared with the SA or LT groups (Figure 3A,B). Moreover, after standardizing against body weight, the thickness of the SMC layer was also 45.5% (*p <* 0.01) and 43.2% (*p <* 0.01) higher, respectively, in the IBA group as compared with the SA or LT group (Figure 3C). The lumen area was increased by 49.7% (*p <* 0.01) in the LT group (Figure 3A,D), and was elevated by 45.6% (*p <* 0.01) after standardizing by body weight in the LT group as compared with the SA group (Figure 3E). However, the lumen area was reduced by 43% (*p <* 0.01, Figure 3D) and by 35.1% (*p <* 0.01, Figure 3E) after standardizing by body weight in the IBA group as compared with the LT group.

### 2.4. Changes in Expression and Localization of Proteins Associated with Phenotypic Switching in SMCs of the Thoracic Aorta of Ground Squirrels

Compared with the SA ground squirrels, the protein expression of the SMC contractile marker protein α-SMA in the thoracic aorta did not change significantly in either LT or IBA groups (Figure 4A,B), and the protein expression of SM22α was significantly reduced in both LT and IBA groups by 50% (*p <* 0.01) and 41.3% (*p <* 0.05), respectively (Figure 4A,C). The synthetic marker protein VIM was significantly increased in SMCs by 114.9% (*p <* 0.01) and 112.9% (*p <* 0.01), respectively, in LT and IBA groups as compared with the SA group (Figure 4A,D). However, the synthetic marker protein OPN showed no change in the LT group but increased by 91.8% (*p <* 0.05) in the IBA group (Figure 4A,E). The proliferation-related PCNA increased by 147% (*p <* 0.01) in the LT group as compared with the SA group (Figure 4A,F). However, PCNA expression was reduced by 34.5% (*p <* 0.05) in the IBA group (Figure 4A,F) as compared with the LT group.

Immunofluorescence results show that the contractile marker protein α-SMA exhibited a significant nuclear translocation in the IBA group as compared with the SA group (Figure 5), and SM22α also exhibited a significant nuclear translocation in both the LT and IBA groups (Figure 6). The synthetic marker protein VIM was significantly localized and closely arranged in the intima of the thoracic aorta in both LT and IBA groups (Figure 7). The localization of both OPN and PCNA did not change significantly between three groups (Figure 8).

### 2.5. Changes in Endothelial Permeability and the Expression Level and Localization of eNOS in the Thoracic Aorta of Ground Squirrels

Serum levels of the endothelial permeability markers HS and Sy-1 were significantly reduced by 41.1% (*p <* 0.01) and 35.1% (*p <* 0.05) in the LT group as compared with the SA group, respectively (Figure 9A,B). HS levels rose again by 43.4% (*p <* 0.01) in the IBA group as compared with the LT group but were not significantly different from either the SA or LT values (Figure 9A). The protein expression of eNOS did not change significantly in the LT or IBA groups as compared with the SA group (Figure 10A,B), and, similarly, the immunofluorescence showed no significant change in eNOS localization in either the LT or IBA groups (Figure 10C).

## 3. Discussion

### 3.1. The Thoracic Aorta and Serum Oxidative Stress Levels of Ground Squirrels Change Periodically during Torpor–Arousal

Increases in MDA and H_2_O_2_ levels in an ischemic–hypoxic state imply increased oxidative stress [53]. In the present study, the levels of MDA and H_2_O_2_ in the thoracic aorta of ground squirrels, representing ROS levels, were significantly elevated in the LT group as compared with the SA group (Figure 1A,B). This is consistent with the findings of significantly elevated serum MDA [13] and H_2_O_2_ levels in the lungs of rats following chronic hypoxic exposure [11]. Similarly, previous studies have reported a significant increase in MDA levels in the plasma of hibernating black bears [54]. Moreover, in addition to a significant increase in ROS production, ROS also significantly increased during arousal in the intestine of the thirteen-lined ground squirrels during torpor [55,56], which is consistent with our findings. In the present study, although the MDA levels in the thoracic aorta were reduced during IBA in ground squirrels (Figure 1A), the H_2_O_2_ level remained at the same high level as in the LT group (Figure 1B), which led to a weaker increase in ROS in the IBA group as compared with the LT group, but this was still elevated as compared with the SA group.

SOD activity was significantly reduced in rat thoracic aortic homogenates after hypoxic–ischemia [21]. SOD activity in the hearts of hibernating ground squirrels (*Citellus citellus*) was the same during the arousal period and the summer-active period [57]. These data are consistent with our results for SOD in serum, which was significantly lower in the LT group than in the SA group and significantly higher in the IBA group than in LT ground squirrels (Figure 1D). However, the levels of SOD activity and protein expression in the heart, brain and liver were similar in torpor and active states of the thirteen-lined ground squirrels [58,59], which is inconsistent with our results. This may be due to the preferential allocation of serum SOD to vital tissues and antioxidant effects after torpor ischemia–hypoxia in ground squirrels, resulting in decreased serum SOD levels in that time period. Surprisingly, the total antioxidant capacity (TAC) of the thoracic aorta was maintained under both LT and IBA (Figure 1C), and previous results from our group also show that the TAC in the heart, liver, brain, kidney and serum of the ground squirrel was significantly higher in the IBA group [53], which may be due to the sensitivity of the thoracic aorta to oxidative stress. In conclusion, it seems reasonable that an appropriate reduction in ROS levels and an increase in antioxidant capacity during IBA may enhance tissue protection over the torpor–arousal cycle in ground squirrels.

### 3.2. Serum Inflammation Levels in the Thoracic Aorta of Ground Squirrels Change Periodically during Torpor–Arousal

Previous studies have shown that Arctic ground squirrels experience suppression of immune responses during torpor [60]. Our results show that serum levels of the pro-inflammatory factors IL-1β and TNFα were significantly elevated during torpor in the ground squirrels as compared with the SA group (Figure 2A,B). This is the same sign of inflammation that is exhibited by hibernating brown bears during torpor [48], except that in brown bears it is manifested by elevated serum levels of haptoglobin and β-macroglobulin [61]. Moreover, the levels of IL-1β and TNFα in placental tissues are increased and the relative anti-inflammatory factor IL-10 is reduced after placental hypoxia in humans [62]; similarly, the present study also found a reduction in IL-10 level during torpor in ground squirrels (Figure 2C). There is also evidence indicating that markers of inflammation-related damage are increased in some hibernating tissues during torpor [63,64,65], which may be reversible upon arousal [39,40]. Similarly, the present study found a reduction in IL-1β during IBA almost equal to the level of the SA group in the serum of ground squirrels (Figure 2A). This is consistent with Arctic ground squirrels experiencing a reduction in systemic inflammation during arousal [66]. Although identical to the findings in bats [67], serum TNFα during IBA in ground squirrels was maintained at a high level comparable to that of the LT group (Figure 2B). This may be related to the timing of arousal, since studies on the brains of hibernating Syrian hamsters found an upregulation of pro-inflammatory factor expression during the early arousal period (90 min) and a recovery after 8 h of arousal [43]. By contrast to changes in pro-inflammatory factors, serum levels of the anti-inflammatory factor IL-10 were elevated to the SA level during IBA in ground squirrels (Figure 2C), which may be related to the recovery of inflammatory and immune responses after rapid blood reperfusion during arousal [68,69]. In summary, the present study suggests that elevated pro-inflammatory factors and decreased anti-inflammatory factors predict that ground squirrels may suffer ischemic injury during torpor. However, the decreased level of pro-inflammatory factor IL-1β and the increased level of anti-inflammatory factor IL-10 during IBA suggest that inflammation was significantly alleviated during the IBA period and that ischemic injury was gradually reduced in the ground squirrels.

### 3.3. Controllable Phenotypic Switching of SMCs and Functional Changes in Proteins Associated with Phenotypic Switching in the Thoracic Aorta of Ground Squirrels during Torpor–Arousal

Inflammation, oxidative stress and hemodynamics are key factors in the process of vascular remodeling [15]. These physiological changes caused by ischemia–hypoxia–reperfusion are also sufficient to induce phenotypic switching of VSMCs [25]. In the present study, the thickness of the SMC layer in the thoracic aorta was significantly increased in the IBA group as compared with the SA group of ground squirrels (Figure 3A,B). The lumen area was increased significantly in the LT group and then decreased back to the SA level in the IBA group (Figure 3A,D). Besides, our results were not affected by body weight (Figure 3C,E) [70]. This phenomenon has also been found in other hibernating mammals, such as hibernating black bears, which have thicker mid-membranes and significantly more SMCs in the aortic arch during torpor [54]. Twenty-eight days of low-pressure hypoxia resulted in thickened pulmonary artery walls and a narrowed lumen in rats [22], but the period of arterial wall thickening in these studies seems to be slightly different from ours, which may be a delayed response to hibernation physiological changes in the ground squirrels themselves. In addition, compared with the SA group, the SMC contractile marker protein SM22α in the thoracic aorta showed a significantly lower protein expression in both the LT and IBA ground squirrels (Figure 4A,C). By contrast, the SMC synthetic marker protein VIM showed a significantly higher protein expression (Figure 4A,D), and the synthetic marker protein OPN showed elevated protein expression only in the IBA group (Figure 4A,E). Furthermore, expression of the proliferation-associated PCNA was significantly elevated in the LT group (Figure 4A,F), which suggests that a significant phenotypic switch occurred in the thoracic aorta of ground squirrels during torpor. Similar to studies on non-hibernators, SM22α protein expression decreased significantly, and PCNA and OPN proteins’ expression increased significantly after 48 h of hypoxia in rat pulmonary artery SMCs [23]. However, based on the recovery of PCNA protein expression in the IBA group to the level of the SA group (Figure 4A,F), we speculate that phenotypic switching in the thoracic aorta of the ground squirrels was reduced during the IBA period.

Compared with the SA group, immunofluorescent results show that α-SMA exhibited a significant nuclear translocation during IBA (Figure 5), as did SM22α in the LT and IBA groups (Figure 6). It has been found that SM22α has a significant negative correlation with the inflammatory response of the vascular wall and that increased expression of SM22α can be involved in regulating vascular wall inflammation by inhibiting the proinflammatory NF-κB pathway [71], and α-SMA can also inhibit the NF-kB pathway [72]. Therefore, we speculate that the nuclear translocation of α-SMA and SM22α might be aimed at better regulating the level of inflammation in the nucleus and thus protecting the genetic material. However, the timing of nuclear translocation is inconsistent between α-SMA and SM22α, reflecting a temporal variability in the effects of the two proteins. In addition, VIM might be involved in the abnormal endothelial barrier changes in the thoracic aorta of squirrels in the LT and IBA groups (Figure 7), because it has been reported that the absence of VIM proteins in abnormal endothelial barrier function leads to impaired leukocyte extravasation [73]. Our findings below also suggest that the endothelial permeability of the thoracic aorta was significantly reduced during LT and IBA. In conclusion, the SMCs of the thoracic aorta exhibited significant phenotypic switching during LT and IBA, but the intensity of the phenotypic switching was significantly reduced during IBA. The results also suggest that the functions of α-SMA, SM22α and VIM proteins were altered during the torpor–arousal cycle.

### 3.4. Periodic Change in Endothelial Permeability and Maintenance of Endothelial Function in the Thoracic Aorta of Ground Squirrels during Torpor–Arousal

Both SMCs and ECs are involved in vascular remodeling after ischemia in organisms [74]. The glycocalyx is a general term for the polymorphic glycoproteins that grow in the apical membrane of vascular ECs [75] and regulate vascular permeability [29]. By contrast, the degradation of the glycocalyx can be demonstrated by elevated levels of its main components, HS and Sy-1 [75]. Ischemia–reperfusion has been shown to lead to glycocalyx degradation via the production of ROS and proinflammatory factors [28,31,32]. Surprisingly, in the present study, serum levels of both HS and Sy-1 were significantly lower in ground squirrels during LT as compared with the SA group (Figure 9A,B). From this, we speculate that the glycocalyx did not undergo degradation during torpor-induced ischemia, but instead promoted reduced endothelial permeability to prevent injury due to adverse infiltration of intra- and extravascular factors [76]. The level of HS in the serum was elevated during IBA in ground squirrels to a level comparable to that of the SA group (Figure 9A) which is evidence of restored permeability. By contrast, in a non-hibernating porcine model, reperfusion exacerbated glycocalyx shedding, leading to elevated HS levels in serum [30].

Ischemia–reperfusion also triggers vascular endothelial dysfunction due to increased ROS production [14,77,78]. It has been reported that the renal arteries of Syrian hamsters show a strong labeling of eNOS throughout the EC cytoplasm during early hibernation arousal (2 h) as compared to controls, and its localization was found to shift from the cytoplasm to the Golgi apparatus during torpor [79]. By contrast, eNOS protein expression in the thoracic aorta did not change significantly in the LT and IBA groups of ground squirrels as compared with the SA group (Figure 10A,B), potentially because the SMC function was maintained during torpor and thus regulated endothelial function [51]. Moreover, immunofluorescence suggested that the localization of eNOS in the thoracic aorta did not change significantly in the LT and IBA groups (Figure 10C). In summary, endothelial permeability was reduced in the thoracic aorta of torpid squirrels in order to maintain a balance of intra- and extravascular factors. The level of permeability gradually recovered during IBA when the level of HS was increased without affecting the standardized penetration of factors inside and outside the vascular into the outside and inside the vascular, and endothelial function was well-maintained in such changes.

## 4. Materials and Methods

### 4.1. Animal Acquisition and Experimental Groups

Animal collection and experimental procedures were approved by the Animal Care Committee of the Chinese Wildlife Conservation Society (SL-2012-42) and the Ethics Committee of Northwest University. As described by our laboratory previously [80,81], thirty Daurian ground squirrels were captured from the Wei’ nan Plain of Shaanxi Province, China, and returned to the laboratory to be weighed and recorded for raw weight, after which water and standard rat food were provided to all animals ad libitum. The ground squirrels in the SA group were settled into an animal room where temperature was maintained at 18–25 °C and the light was changed daily in synchronization with the local sunrise and sunset. In November, when the ground squirrels entered torpor, they were moved to a cold room maintained at 4–6 °C. The body temperature (*T*_b_) of the squirrels was ascertained by thermal imaging with a visual infrared thermometer (Fluke VT04 Visual IR Thermometer, Washington, DC, USA) and was measured twice daily, usually at 9 AM and 9 PM (12 h interval). Based on our observations, the squirrels did not eat or drink during torpor but aroused periodically for 1–2 days before re-entry into torpor (interbout arousal). Based on these observations, we weight-matched and randomly divided the ground squirrels into three groups (10 animals per group): (1) summer active (SA): active squirrels sampled in July with a *T*_b_ of 36–38 °C; (2) late torpor (LT): squirrels hibernating for two months at 4–6 °C, followed by an arousal and then re-entry into hibernation with sampling after 3–4 days of consistent *T*_b_ at 4–6 °C; and (3) interbout arousal (IBA): squirrels naturally aroused after two months of hibernation that displayed a *T*_b_ of 34–37 °C for less than 12 h before sampling.

### 4.2. Sample Collection

All squirrels were rapidly weighed and then anesthetized with 90 mg/kg sodium pentobarbital. After anesthesia, the abdomen was surgically opened, and a blood sample was taken from the main abdominal vein with a blood collection tube without any chemical reagents followed by quickly dissecting out the thoracic aorta. Aorta weight was quickly recorded before removing an ~2 mm section (always from the same location) that was immersed in 4% paraformaldehyde for fixation for subsequent sectioning. The remaining portion of the aorta was quickly frozen in liquid nitrogen. The collected blood was centrifuged at 10,000 rpm for 5 min after being placed at room temperature for 30 min, and the supernatant was carefully aspirated and then centrifuged at 3000 rpm for 5 min, and then the serum with clean supernatant was carefully aspirated. At the end of this procedure, all animals were euthanized by injection of an overdose of sodium pentobarbital. Subsequently, the thoracic aortic samples soaked in paraformaldehyde were stored at 4 °C, and the serum and liquid-nitrogen-frozen thoracic aortic samples were stored at −80 °C until further processing.

### 4.3. Protein Extraction and Concentration Determination

Total soluble protein extracts were prepared from frozen thoracic aortic samples of ten squirrels from each experimental group (SA, LT, IBA), as described previously [82]. Briefly, frozen tissue samples (~30 mg) were homogenized in RIPA lysis buffer (Heart, Xi’an, China) with 10 μL/mL protease inhibitor mixture (Heart, Xi’an, China) added and then left on ice for 30 min. Samples were centrifuged at 4 °C for 15 min at 15,000 to obtain supernatants. Soluble protein concentrations were determined by the BCA method (Boster, Wuhan, China). Protein supernatants from five animals were then frozen at −80 °C for subsequent experiments, and protein supernatants from the remaining five animals were adjusted to a concentration of 2.0 μg/μL using 1× SDS loading buffer (Boster, Wuhan, China) followed by boiling and then storage at −80 °C until subsequent use.

### 4.4. Quantification of ROS

ROS are extremely short-lived and highly reactive; hence, their exact measurement in tissue samples remains difficult [53,83]. Here, measurements of MDA (a secondary product) and H_2_O_2_ (an important ROS) were used as indicators of ROS levels [84]. MDA and H_2_O_2_ concentrations in protein supernatants of thoracic aortic samples were measured using MDA and H_2_O_2_ assay kits (Nanjing Jiancheng, Nanjing, China), respectively, according to the manufacturer’s protocol. The supernatant from the thoracic aorta protein extract was mixed with the analytical reagent containing thiobarbituric acid (TBA), and the mixture was heated at 100 °C and cooled to room temperature. Then, the mixture was centrifuged at 3000 rpm for 15 min at 4 °C. The absorbance of the supernatant was then measured by spectrophotometry (Shimadzu UV-2550, Kyoto, Japan) at 532 nm, and finally, MDA concentration in the thoracic aorta was determined by comparing its OD532 value with the MDA standard value. H_2_O_2_ reacts with molybdic acid to form stable molybdic acid peroxide compounds, and absorbance was measured spectrophotometrically at 405 nm (Shimadzu, UV-2550, Kyoto, Japan), and finally, the H_2_O_2_ concentration in the thoracic aorta was determined by comparing its OD405 value with the H_2_O_2_ standard value.

### 4.5. Quantification of Antioxidant Capacity

The TAC in thoracic aortic supernatant was measured by a biochemical assay using a TAC assay kit (Nanjing Jiancheng, Nanjing, China) according to the manufacturer’s protocol. The supernatant was mixed with kit reagent at 37 °C for 30 min and then mixed again, and absorbance was measured at 520 nm to calculate the content by formula. SOD levels in serum were measured by ELISA using the SOD assay kit (F3262-B, Fankew, Shanghai, China) according to the manufacturer’s protocol. A double-antibody sandwich method was used to coat the microtiter plate with purified rat SOD to capture antibody, and rat SOD was added sequentially to the coated wells and then combined with horseradish peroxidase (HRP)-labeled detection antibody, and after thorough washing, the substrate was added for color development. The absorbance was measured at 450 nm using an enzyme standardization instrument to determine the SOD content in serum.

### 4.6. Quantification of Inflammatory Factors

Antibodies were purchased from Fankew, Shanghai, China, for the following proteins (catalogue number in brackets): IL-1β (F2923-B), TNFα (F3056-B), IL-10 (F3071-B) and interleukin 4 (IL-4; F3067-B). A double-antibody sandwich method was used to cover the microtiter plate with purified rat IL-1β capture antibody, and rat IL-1β was added sequentially to the covered microtiter wells and then combined with HRP-labeled detection antibody, which was washed thoroughly and then colored with the substrate. Absorbance was measured at 450 nm with an enzyme marker to determine the content of each protein of interest in the serum. The same method was used for the detection of TNFα, IL-10 and IL-4 in serum.

### 4.7. Western Blots

As described by a previous study [85], 10% SDS-PAGE gels were used to separate thoracic aorta proteins after adding 1× SDS to protein supernatant samples. Electrophoresis was performed at 80 V for 30 min and then 60 min at 120 V. Proteins were then transferred onto polyvinylidene difluoride (0.20 μm pore size, Merck, Darmstadt, Germany) membranes using semi-dry transfer apparatus (Bio-Rad, Hercules, Calif, USA) at 20 V for 5 min, 40 V for 10 min and 60 V for 30 min. Membranes were then treated with 4% PVA-203 (Aladdin, Shanghai, China) in Tris-buffered saline for 10 min [86,87] and then incubated with diluted rabbit primary antibodies as follows: anti-α-SMA (1:5000, Abcam, ab14106, Cambridge, UK), anti-SM22α (1:5000, Abcam, ab32575, Cambridge, UK), anti-**VIM** (1:10,000, Abcam, ab92547,Cambridge,UK), anti-OPN (1:2000, Proteintech, 22952-1-AP, Wuhan, China), anti-PCNA (1:5000, Cell Signaling Technology, D3H8P, Danvers, Mass., USA) or anti-eNOS (1:10,000, Abcam, ab199956, Cambridge, UK), overnight at 4 °C in 0.1% TBST containing 2% polyvinylpyrrolidone (PVP40; Amresco, Houston, Texas, USA). The membranes were then washed three times for 10 min each with 0.1% TBST and continuous agitation followed by incubation with HRP-conjugated anti-rabbit secondary antibody (1:10,000 *v*:*v*, Zhuangzhi, EK020, Xi’an, China) for 2 h at room temperature, followed by three washes (10 min/time) with 0.1% TBST. Fluorescent bands were observed using an enhanced chemiluminescence reagent (Thermo Fisher Scientific, NCI5079, Waltham, MA, USA). Quantitative analysis was performed using ImageJ software. The density of immunoblot bands in each lane was standardized by the total density of total protein bands in the same lane [88,89].

### 4.8. Histological Analysis Using Hematoxylin and Eosin (HE) Staining

Tissues were removed from the paraformaldehyde fixative and dehydrated by a gradient concentration of alcohol. Wax was melted at 65 °C and placed into the embedding frame, and then tissues were placed into the embedding frame and labeled accordingly before the wax solidified and then cooled at −20 °C. The cooled wax pieces were cut into 4 μm slices and spread on warm water at 40 °C, and the slides were picked up and baked in an oven (Labotery, GFL-230, Tianjin, China) at 60 °C. After the water was dried, slides were dewaxed, and slices were stained with hematoxylin (HE) staining solution for 3–5 min, washed with tap water, differentiated with differentiation solution, washed with tap water, returned to blue with blue return solution and rinsed with running water. The sections were dehydrated in gradient alcohol for 5 min each and stained with eosin staining solution for 5 min. After dehydration, the sections were sealed with neutral gum and microscopically examined (Nikon Eclipse E100, Nikon, Japan), and images were acquired and analyzed (Nikon DS-U3, Nikon, Japan). Finally, the smooth muscle layer thickness and lumen area of the thoracic aorta were measured with Image Pro Plus software.

### 4.9. Immunofluorescent Analysis

Wax blocks were made for HE staining. Paraffin sections were dewaxed in water followed by EDTA antigen repair buffer (pH 8.0) for antigen repair. After natural cooling, the slides were placed in PBS (pH 7.4) and washed 3 times for 5 min each time on a decolorization shaker. The sections were lightly shaken dry and incubated for 25 min at room temperature and protected from light in 3% hydrogen peroxide solution after drawing circles around the tissue with a histochemical pen. Primary antibodies were then applied: anti-α-SMA (1:500, Abcam, ab14106, Cambridge, UK), anti-SM22α (1:300, Abcam, ab32575, Cambridge, UK), anti-VIM (1:1000, Servicebio, GB11192, Wuhan, China), anti-OPN (1:50, Proteintech, 22952-1-AP, Wuhan, China), anti-PCNA (1:800, Servicebio, GB11192, Wuhan, China) or anti-eNOS (1:300, Servicebio, GB12086, Wuhan, China). Sections were incubated flat in a wet box at 4 °C overnight, and the slides were washed as before. The sections were shaken dry and covered with secondary antibody Cy3 (1:300, Servicebio, Wuhan,China) in a circle drop by drop and incubated for 50 min at room temperature, and then slides were washed as before. After the slice was slightly shaken and dried, DAPI (Servicebio, G1012, Wuhan, China) staining solution was added dropwise in the circle and incubated for 10 min at room temperature, protected from light, and then slides were washed as before. After the slides were shaken and dried, autofluorescence quencher was added to the circles for 5 min, and the slides were then rinsed under running water for 10 min. Sections were observed under a fluorescence microscope (Nikon Eclipse C1, Nikon, Japan), and images were acquired (Nikon DS-U3, Nikon, Japan).

### 4.10. Quantification of Endothelial Permeability

Antibodies for acetyl HS (F40462-B) and Sy-1 (F40577-B (both from Fankew, Shanghai, China) were used to quantify these proteins by ELISA methodology according to the manufacturer’s protocol, and the assay method was the same as for IL-1β.

### 4.11. Statistical Analyses

All data were analyzed using SPSS 20.0 and expressed as mean ± SEM. One-way analysis of variance (ANOVA) was used to determine significant differences between groups with the Sidak post hoc test. For data showing non-homogeneity, ANOVA–Dunnett’s T3 method was applied. Values of *p* < 0.05 were considered to be statistically significant.

## 5. Conclusions

In summary, a certain degree of ischemic and hypoxic injury occurs in the bodies of ground squirrels during torpor, and the SMCs of the thoracic aorta undergo significant phenotypic switching with endothelial permeability also significantly reduced in torpor. However, after tissue reperfusion (and body temperature) rose again during arousal, ischemic and hypoxic injury was significantly alleviated, the degree of SMC phenotype switching was diminished, and endothelial permeability returned to the SA level. In addition, the TAC and endothelial function of the thoracic aorta were maintained during torpor–arousal cycles. In conclusion, alterations in oxidative stress and inflammation levels over torpor–arousal cycles induced controlled phenotypic switching as well as restoration of endothelial permeability in thoracic aorta SMCs in order to resist ischemia–reperfusion injury in Daurian ground squirrels.

## Figures and Tables

**Figure 1 ijms-23-10248-f001:**
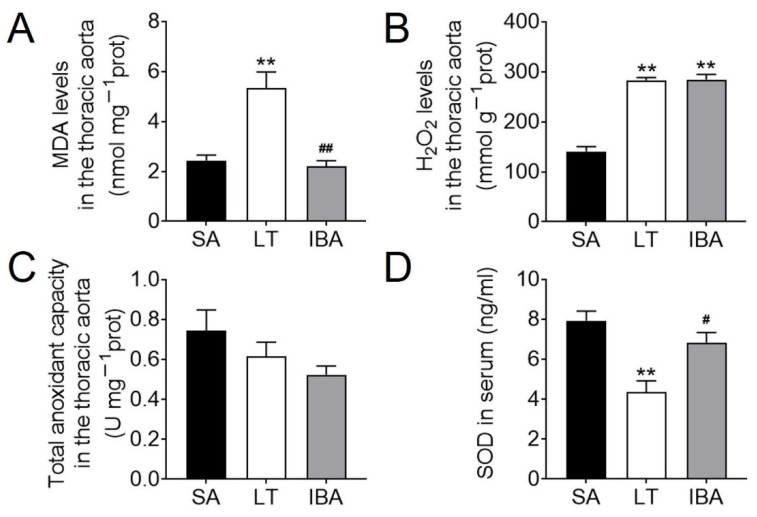
Levels of ROS and antioxidant capacity in the thoracic aorta and serum of ground squirrels. (**A**) MDA content in the thoracic aorta, *n* = 3. (**B**) H_2_O_2_ content in the thoracic aorta, *n* = 5. (**C**) TAC in the thoracic aorta, *n* = 5. (**D**) SOD content in serum, *n* = 6. Samples are SA: summer active, LT: late torpor, and IBA: interbout arousal. Data are mean ± SEM. Significant differences between SA samples are denoted as follows: ** *p <* 0.01, whereas significant differences in IBA as compared with LT are denoted by # *p <* 0.05 or ## *p <* 0.01.

**Figure 2 ijms-23-10248-f002:**
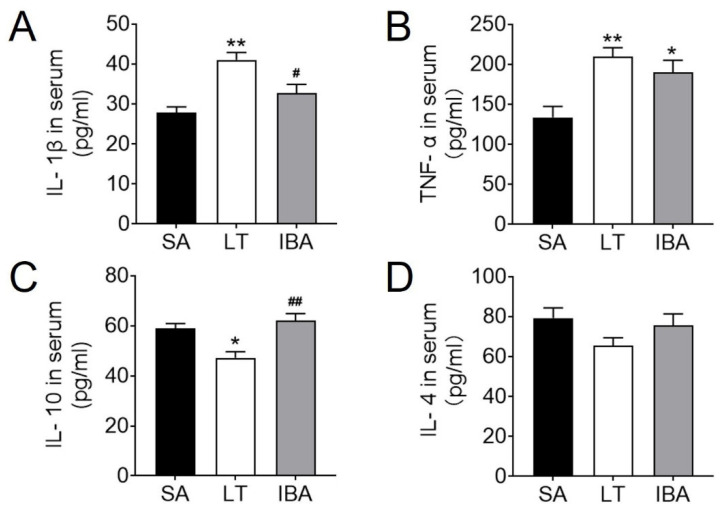
Serum levels of inflammatory factors in ground squirrels. (**A**) Serum levels of pro-inflammatory factor IL-1β. (**B**) Serum levels of pro-inflammatory factor TNF-α. (**C**) Serum levels of anti-inflammatory factor IL-10. (**D**) Serum levels of anti-inflammatory factor IL-4. SA: summer active, LT: late torpor, IBA: interbout arousal. Data are mean ± SEM, *n* = 6. Significant differences between SA conditions are denoted as follows: * *p <* 0.05, ** *p <* 0.01, whereas significant differences in IBA as compared with LT are denoted by # *p <* 0.05, ## *p <* 0.01.

**Figure 3 ijms-23-10248-f003:**
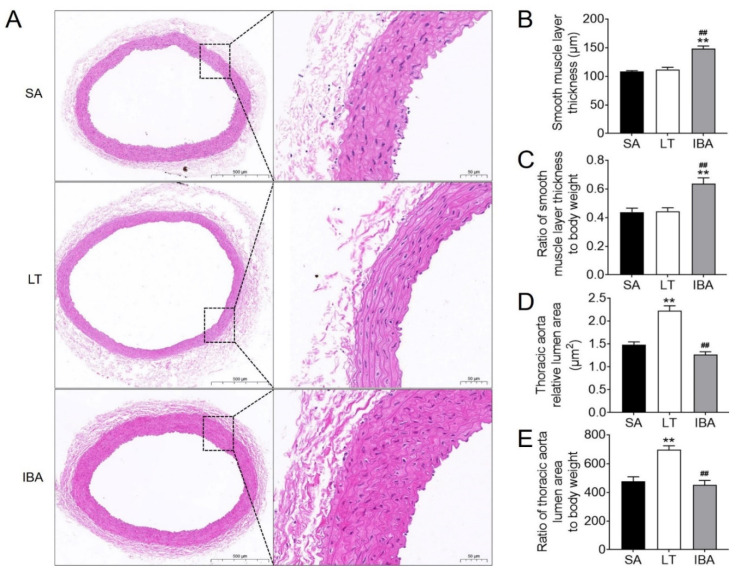
HE staining of the thoracic aorta in ground squirrels. (**A**) Typical images of HE staining. The scale in the left column is 500 μm and the scale in right column is 50 μm. (**B**) Smooth muscle layer thickness was measured at three random locations in each thoracic aorta. (**C**) The ratio of the mean value of smooth muscle layer thickness to body weight was determined for three random locations in each thoracic aorta. (**D**) Lumen area of the thoracic aorta. (**E**) Ratio of lumen area of thoracic aorta to body weight. SA: summer active, LT: late torpor, IBA: interbout arousal. Mean ± SEM. *n* = 6. Significant a difference from SA conditions is denoted as ** *p <* 0.01, whereas a significant difference in IBA as compared with LT is denoted by ## *p <* 0.01.

**Figure 4 ijms-23-10248-f004:**
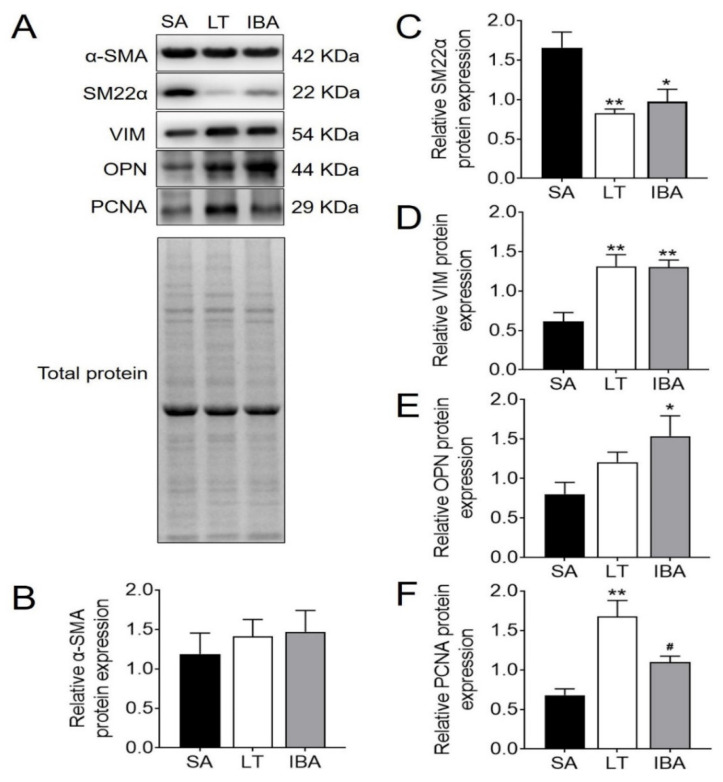
Expression levels of phenotype-switching related proteins in thoracic aortic SMCs. (**A**) Typical Western blot images of phenotype-switching related proteins in thoracic aortic SMCs. (**B**) Protein expression statistics of α-SMA in thoracic aortic smooth muscle. (**C**) Protein expression statistics of SM22α in thoracic aortic smooth muscle. (**D**) Protein expression statistics of VIM in thoracic aortic smooth muscle. (**E**) Protein expression statistics of OPN in thoracic aortic smooth muscle. (**F**) Protein expression statistics of PCNA in thoracic aortic smooth muscle. SA: summer active, LT: late torpor, IBA: interbout arousal. Mean ± SEM. *n* = 5. Significant differences between SA conditions are denoted as follows: * *p <* 0.05, ** *p <* 0.01, whereas significant differences in IBA as compared with LT are denoted by # *p <* 0.05.

**Figure 5 ijms-23-10248-f005:**
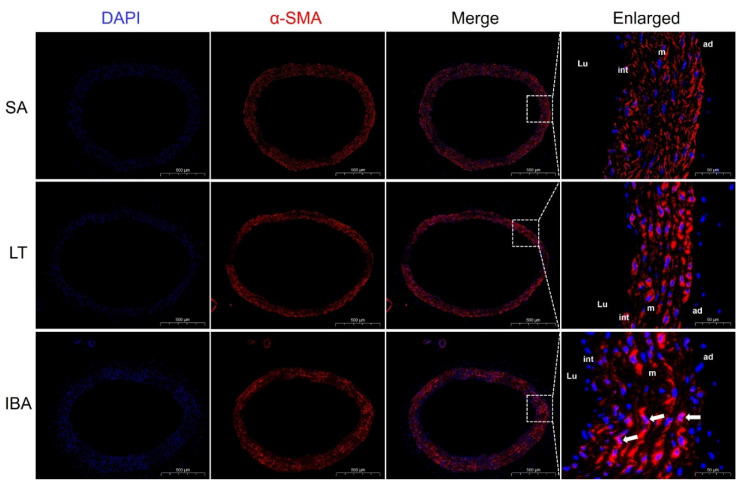
Immunofluorescence staining of α-SMA in the smooth muscle of thoracic aorta. DAPI (blue) labels the nucleus, and CY3 (red) labels α-SMA. The scale in left column = 500 μm. The scale in right column = 50 μm. The white arrows are nuclear translocation changes. SA: summer active, LT: late torpor, IBA: interbout arousal, Lu: lumen, int: intima, m: media, ad: adventitia.

**Figure 6 ijms-23-10248-f006:**
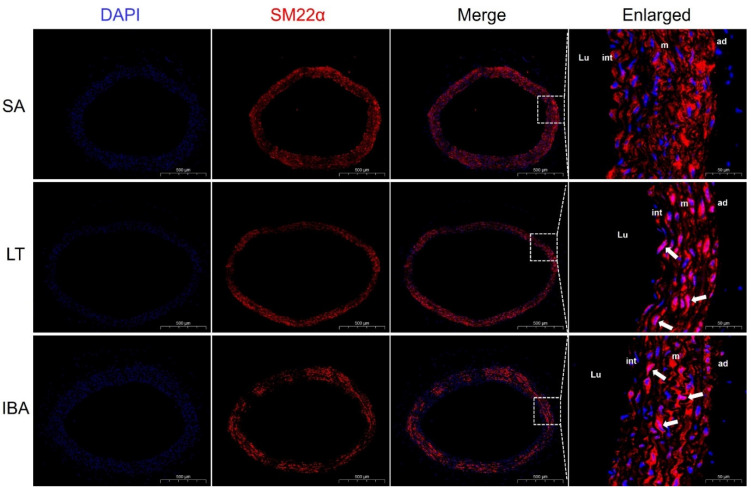
Immunofluorescence staining of SM22α in the thoracic aorta of ground squirrels. DAPI (blue) labels the nucleus, and CY3 (red) labels nuclear SM22α. The scale in the left column = 500 μm and in the right column = 50 μm. White arrows are nuclear translocation changes. SA: summer active, LT: late torpor, IBA: interbout arousal, Lu: lumen, int: intima, m: media, ad: adventitia.

**Figure 7 ijms-23-10248-f007:**
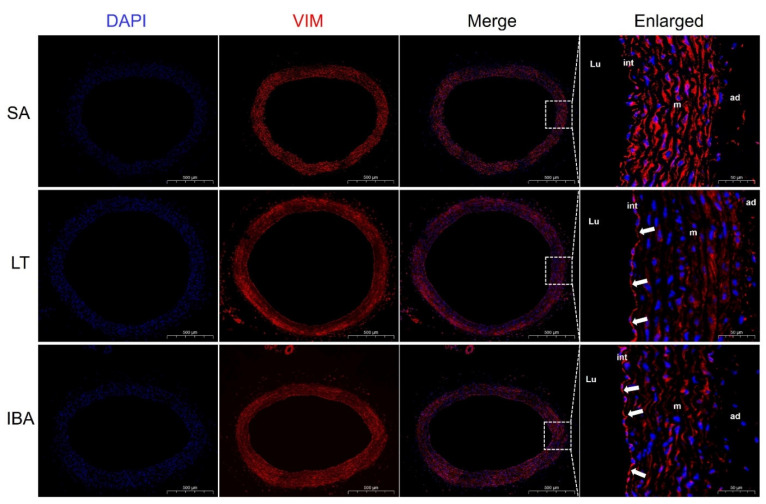
Immunofluorescence staining of VIM in thoracic aorta. DAPI (blue) labels the nucleus, and CY3 (red) labels VIM. The scale in left column = 500 μm. The scale in right column = 50 μm. Changes in endothelial localization are shown by white arrows. SA: summer active, LT: late torpor, IBA: interbout arousal, Lu: lumen, int: intima, m: media, ad: adventitia.

**Figure 8 ijms-23-10248-f008:**
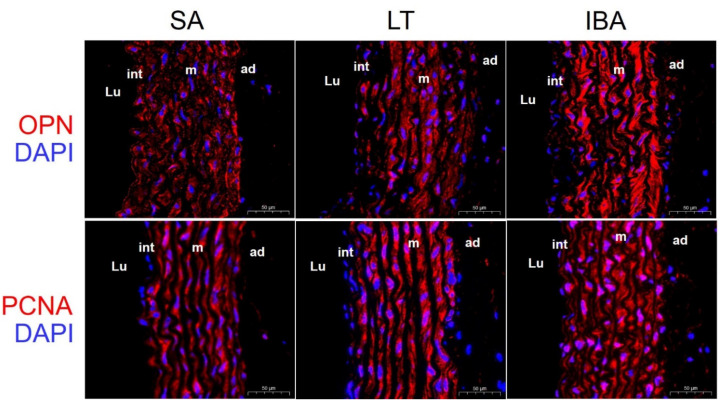
Immunofluorescence staining of OPN and PCNA in thoracic aorta. DAPI (blue) labels the nucleus, and CY3 (red) labels OPN and PCNA. The scale bar is 50 μm. SA: summer active, LT: late torpor, IBA: interbout arousal, Lu: lumen, int: intima, m: media, ad: adventitia.

**Figure 9 ijms-23-10248-f009:**
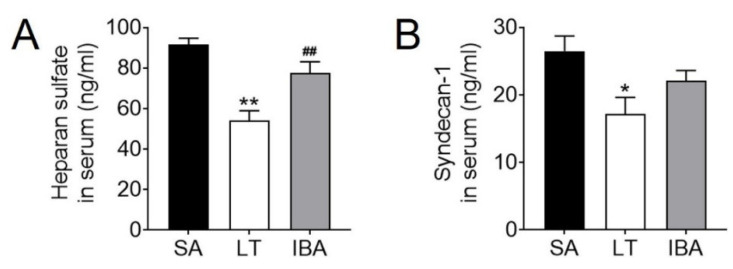
Serum endothelial permeability marker levels in ground squirrels. (**A**) Serum levels of HS. (**B**) Serum levels of Sy-1. SA: summer active, LT: late torpor, IBA: interbout arousal. Mean ± SEM. *n* = 6. Significant differences between SA conditions are denoted as follows: * *p <* 0.05, ** *p <* 0.01, whereas significant differences in IBA as compared with LT are denoted by ## *p <* 0.01.

**Figure 10 ijms-23-10248-f010:**
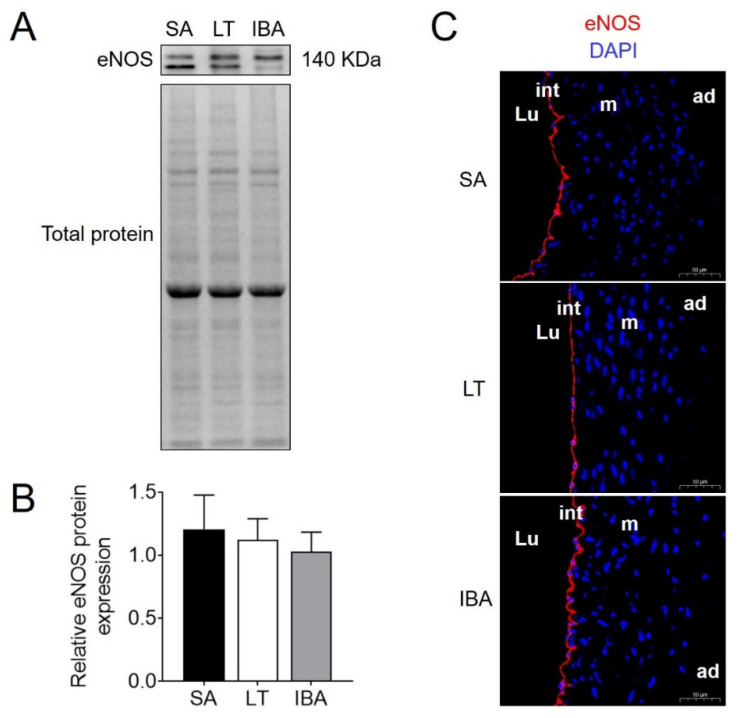
The protein expression level and localization of eNOS in thoracic aorta. (**A**) Typical Western blot image of eNOS protein. (**B**) Statistical image of protein expression of eNOS in thoracic aorta from SA, LT and IBA squirrels, mean ± SEM, *n* = 5. (**C**) Immunofluorescence staining of eNOS in thoracic aorta, scale bar = 50 μm. DAPI (blue) labels the nucleus, and CY3 (red) labels the eNOS protein. SA: summer active, LT: late torpor, IBA: interbout arousal, Lu: lumen, int: intima, m: media, ad: adventitia.

## Data Availability

The data presented in this study are available on request from the corresponding author.

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
