# Peer review of "The Protective Effects on Ischemia–Reperfusion Injury Mechanisms of the Thoracic Aorta in Daurian Ground Squirrels (Spermophilus dauricus) over the Torpor–Arousal Cycle of Hibernation"

_ijms, 2022, doi:10.3390/ijms231810248_

Round 1

Reviewer 1 Report

The manuscript is well-written, the topic is interested and provide important information about the protective mechanisms on ischemia-reperfusion injury in Daurian ground squirrels.

Minor comments:

Many studies prove a strong correlation between the sex of the experimental animals and the susceptibility and progression of ischemia-reperfusion injury. In rodent studies, it is well-established that the cardiovascular system of female animals are less vulnerable to inflammatory and oxidative damages. Beside the sex factor, the age may also predispose to the cardiac and vascular damages. How important are these important factors in this experiment?

Please, check the format (font style) and correct the grammatical errors of the MS.

Author Response

Review 1

  1. The manuscript is well-written, the topic is interested and provide important information about the protective mechanisms on ischemia-reperfusion injury in Daurian ground squirrels.

Answer: Thank you very much for your approval, In the future, we will continue to carry out related experiments on the basis of the present, hoping to further expand the research and application scope of Daurian ground squirrels.

  1. Many studies prove a strong correlation between the sex of the experimental animals and the susceptibility and progression of ischemia-reperfusion injury. In rodent studies, it is well-established that the cardiovascular system of female animals are less vulnerable to inflammatory and oxidative damages. Beside the sex factor, the age may also predispose to the cardiac and vascular damages. How important are these important factors in this experiment?

Answer: We really appreciate the reviewer for highlighting this point. We are very sorry that we did not give a more detailed description of the sex and age of Daurian ground squirrels.

First of all, we strongly agree that the sex of the Daurian ground squirrels used can lead to different results. In fact, for wild animals, capture is more random and most of the time there are more females than males, but when designing this experiment, we tried to design the male to female ratio to be 1:1 or close to 1:1 as much as possible, on the basis of individual experiments with a large difference between male and female ratios we ensured that the sample size was at the level of n ≥ 5 to avoid experimental errors due to sex, for which we have made a table (table 1) for you to review again for the sex of the ground squirrels used in each experiment. In addition, we are always looking for ways to improve the sex uncertainty caused by field capture, so that later studies can be more rigorous.

Table 1: Sex of experimental Daurian ground squirrels

Experimental methods

Indicator name

Male

Female

Biochemical experiments

malondialdehyde (MDA)

5

4

hydrogen peroxide (H2O2)

5

10

total antioxidant capacity (TAC)

8

7

ELISA

tumor necrosis factorα (TNFα), interleukin-1β (IL-1β), interleukin-10 (IL-10), interleukin-4 (IL-4), heparan sulfate (HS), syndecan-1 (Sy-1)

8

10

Western blots

α-smooth muscle actin (α-SMA)

6

9

smooth muscle 22α (SM22α)

5

10

vimentin (VIM)

9

6

osteopontin (OPN)

8

7

proliferating cell nuclear antigen (PCNA)

7

8

endothelial nitric oxide synthase (eNOS)

6

9

HE

thickness of the SMC layer(ratio to body weight)

7

11

thickness of the SMC layer

10

8

lumen area(ratio to body weight)

9

9

lumen area

10

8

IF

αSMA, SM22α, VIM, OPN, PCNA

13

11

Secondly, we regretted that we cannot accurately know the exact age of the ground squirrels caught in the wild. However, in order to improve the accuracy of the experiment, we will still use some standards to determine the approximate age of the ground squirrels. Specifically, we chose to capture young and adult ground squirrels. We determined the age of the ground squirrels by observing their teeth and claws. The incisors of adult ground squirrels are white, short and neatly arranged. With the increase of age, the incisors of aging ground squirrels are dark yellow, thick and long, and unevenly arranged, and the edges of the teeth are broken. In addition, the claws of adult ground squirrels are short and straight, hidden in the hair of feet. With the increase of age, the claws are exposed beyond the hair of feet, and the tips of claws are bent. Based on these characteristics, aging ground squirrels were excluded by us. Similarly, based on body size and weight, sub-adults were also excluded.

  1. Please, check the format (font style) and correct the grammatical errors of the MS.

Answer: Sincerely thank you for your constructive suggestion. We have carefully proofread the language to ensure that there are no errors as much as possible.

Reviewer 2 Report

General comments to Authors

This study investigated the changes in the thoracic aorta and serum in summer active (SA), late torpor (LT) and interbout arousal (IBA) Daurian ground squirrels. The data presented show that protective mechanisms involved in the ability to resist to vascular ischemia-reperfusion injury during torpor of Daurian ground squirrels include the decrease in reactive oxygen species (ROS) and pro inflammatory factors and the increase of SOD and anti-inflammatory factors during the IBA period, leading to controlled phenotypic switching of thoracic aortic SMCs and restoration of endothelial permeability.

I think this is an interesting study, giving a further contribution to understand the mechanisms of vascular resistance to ischemia-reperfusion injury.

Author Response

Review 2

This study investigated the changes in the thoracic aorta and serum in summer active (SA), late torpor (LT) and interbout arousal (IBA) Daurian ground squirrels. The data presented show that protective mechanisms involved in the ability to resist to vascular ischemia-reperfusion injury during torpor of Daurian ground squirrels include the decrease in reactive oxygen species (ROS) and pro inflammatory factors and the increase of SOD and anti-inflammatory factors during the IBA period, leading to controlled phenotypic switching of thoracic aortic SMCs and restoration of endothelial permeability. I think this is an interesting study, giving a further contribution to understand the mechanisms of vascular resistance to ischemia-reperfusion injury.

Answer: Thank you very much for your affirmation of our work. Daurian ground squirrel is periodic fattening and hibernation are very interesting issues. Our laboratory has been conducting research on Daurian ground squirrels for a long time. In the future, we will continue to carry out related experiments on the basis of the present, hoping to further expand the research and application scope of Daurian ground squirrels.
